# Segmentation of Plantar Foot Thermal Images Using Prior Information

**DOI:** 10.3390/s22103835

**Published:** 2022-05-18

**Authors:** Asma Bougrine, Rachid Harba, Raphael Canals, Roger Ledee, Meryem Jabloun, Alain Villeneuve

**Affiliations:** 1Multidisciplinary Research Laboratory in Systems Engineering, Mechanics and Energy (PRISME), University of Orleans, 12 rue de Blois, 45067 Orleans, France; rachid.harba@univ-orleans.fr (R.H.); raphael.canals@univ-orleans.fr (R.C.); roger.ledee@univ-orleans.fr (R.L.); meryem.jabloun@univ-orleans.fr (M.J.); 2The Diabetic Foot Service, Regional Hospital of Orleans, 45100 Orleans, France; alain.villeneuve@chr-orleans.fr

**Keywords:** prior-shape-based segmentation, active contours, plantar foot thermal images, infrared camera, diabetic foot, hyperthermia

## Abstract

Diabetic foot (DF) complications are associated with temperature variations. The occurrence of DF ulceration could be reduced by using a contactless thermal camera. The aim of our study is to provide a decision support tool for the prevention of DF ulcers. Thus, the segmentation of the plantar foot in thermal images is a challenging step for a non-constraining acquisition protocol. This paper presents a new segmentation method for plantar foot thermal images. This method is designed to include five pieces of prior information regarding the aforementioned images. First, a new energy term is added to the snake of Kass et al. in order to force its curvature to match that of the prior shape, which has a known form. Second, we defined the initial contour as the downsized prior-shape contour, which is placed inside the plantar foot surface in a vertical orientation. This choice makes the snake avoid strong false boundaries present outside the plantar region when evolving. As a result, the snake produces a smooth contour that rapidly converges to the true boundaries of the foot. The proposed method is compared to two classical prior-shape snake methods, that of Ahmed et al. and that of Chen et al. A database of 50 plantar foot thermal images was processed. The results show that the proposed method outperforms the previous two methods with a root-mean-square error of 5.12 pixels and a dice similarity coefficient of 94%. The segmentation of the plantar foot regions in the thermal images helped us to assess the point-to-point temperature differences between the two feet in order to detect hyperthermia regions. The presence of such regions is the pre-sign of ulcers in the diabetic foot. Furthermore, our method was applied to hyperthermia detection to illustrate the promising potential of thermography in the case of the diabetic foot. Associated with a friendly acquisition protocol, the proposed segmentation method is the first step for a future mobile smartphone-based plantar foot thermal analysis for diabetic foot patients.

## 1. Introduction

Diabetes is a growing health problem worldwide. In 2014, 422 million people suffered from this disease, and 500 million are expected in 2030. Diabetic Foot (DF) is a common disease among diabetic patients, leading to foot ulceration, which is the primary reason for most foot amputations. According to diabetes experts, ulcer occurrence can further be reduced by including thermal information, which is not yet taken into account in the clinical routine. DF is associated with temperature variations, which can be identified using an infrared camera [1,2,3].

Thus, the use of an infrared camera is fast, painless, inexpensive, non-invasive, contactless and allows visualization of the temperature distribution with a precise quantification. Interest in thermography for DF images is rapidly growing. Armstrong et al. [4] demonstrated that the occurrence of a foot ulcer is associated with foot hyperthermia. Hyperthermia is defined as a temperature difference greater than 2.2 °C between a foot region and the same region on the contralateral foot.

Van Netten et al. [5,6], Liu et al. [7], Kaabouch et al. [8,9], Vilcahuaman et al. [10,11] presented an asymmetric analysis based on hyperthermia identification to detect pre-signs of ulceration. Peregrina-Barreto et al. [12] studied the temperature difference and the distribution within the four angiosome regions of the foot. This work proposed a hot spot estimator to detect abnormal areas with high temperature. Other studies focused on the skin temperature distribution and the identification of healthy skin compared to damaged skin.

In 1991, Chan et al. [13] described the temperature distribution of the healthy foot as symmetric butterfly patterns. The highest temperature is at the arch and the lowest appears at the toes. Nagase et al. [14] identified this pattern in 46% of their database and characterized seven other normal patterns. The authors also characterized 18 abnormal patterns in diabetic foot subjects. More recently, Hernandez-Contreras et al. [15,16,17,18] proposed a classification system to identify changes in the thermal distribution of patients suffering from DF compared to healthy subjects.

In most of the above studies, the acquisition protocol and the data processing are as follows. After lying down, the patient has to place their feet in a special device that hides all other thermal sources, except those of the plantar foot [7,8,10,19,20,21]. Images are taken with a thermal camera placed on a tripod, and the data is transferred from the camera to a computer for processing. This acquisition protocol ensures that the background of the plantar foot is homogeneous and that only the plantar foot surfaces are present in the infrared image.

The segmentation of the foot, in this case, is an easy task. For example, in [15,17], a temperature threshold was applied to extract the plantar region from the homogeneous background. In [8,9,22], the authors used a genetic algorithm for the segmentation. In the same way, the authors in [10,11] used an active contour without edges [23].

On the other hand, the use of this constraining protocol (complex acquisition protocol and transferring data into a computer for processing) is not convenient for the medical staff nor for the patient. As a result, the thermal analysis of DF is difficult to include in routine clinical work. To do this, we propose an overall friendly and mobile protocol (acquisition, transfer and processing) for plantar foot thermal images. Images are taken freehand with a smartphone equipped with a dedicated thermal camera. The full automatic processing of the data will be performed immediately in the smartphone itself.

The first step of this new protocol is a fully automatic segmentation of the plantar foot surface. Images are acquired without a special device. The automatic segmentation of such images is a difficult task because of the occurrence of thermal sources other than those of the plantar foot. The presence of these thermal sources leads to the absence of foot boundaries in specific areas of the image, resulting in false boundaries (see Figure 1). Classical segmentation methods, such as [23,24,25], methods with level-set formulation [26,27,28] and graph cut-based methods [29] were tested on our specific images.

These methods failed to segment the plantar foot region in the thermal images [30]. To address this segmentation challenge, several approaches are possible, such as deep Convolutional Neural Networks (CNNs) [31] and prior-shape-based segmentation methods. CNN will not be considered here because these methods require a large database to train the network, which is not yet available. Prior-shape active-contour methods, however, can be a solution when the edges of the object to segment are weak or absent in the image. Several prior-shape-based active-contour methods have been developed in the literature.

Cootes et al. [32] extended the geodetic active-contours model to developed a statistical-point-distribution model, called the active shape model (ASM), in order to form the object shape. A number of markers were located on training images of the object, and the average positions of the markers and their patterns of variation were used to establish the ASM.

Yang et al. [33] proposed a method based on the statistical neighbor prior information to segment multiple objects simultaneously. They defined a MAP estimation framework combining both the neighbor prior information provided by neighboring objects as well as the image gray level information in a level-set active-contour formulation.

Tsai et al. [34] presented an approach for statistical pattern classification based on the “expectation-maximization (EM)” algorithm and the integration of prior shapes with the level-set method. Leventon et al., in [35], presented a probabilistic approach. The model is based on Principal Component Analysis, the estimation of a maximum a posteriori position and the shape of the object in a level-set formulation. A Bayesian approach combined with multiple shape priors was also proposed by Chang et al. [36].

Another Bayesian approach [37] was described by Yeo et al. in 2014. The statistical shape information was incorporated using a nonparametric shape-density distribution in a variational level-set model. The method of Chen et al. [38] proposed finding transformation parameters related to the scale, rotation and translation to match the prior-shape contour. A Fourier-based prior-shape method was presented by Ahmed et al. [39] in which the shape matching was performed in the Fourier space.

In the present paper, we propose a new segmentation method for DF plantar thermal images, which considers all the prior information. The following information will be included in the new segmentation procedure:The plantar foot contour to be found is a single closed contour. Since we divide the acquired thermal image containing both feet into two, we segment each plantar foot image separately.The average shape of the plantar foot is known by taking an average contour of several plantar feet.The plantar foot contour and the average shape contour present both soft curvatures, and there is no strong curvature variations in the foot contour (assimilated more or less to an oval shape).No strong edges are present inside the plantar foot region, while strong ones appear elsewhere in the image where the thermal information of the rest of the body is more prominent.The orientation of the foot is known: as we ask the participants during the acquisition to keep the foot orientation vertical or quasi-vertical).

As we are expecting a single closed contour, a snake-based method is preferred. In a second step, a prior-shape-based snake is chosen to include the remaining prior information. The prior-shape-based snake includes an extra energy term that forces the moving curve to match the prior-shape curve.

Based on the snake of Kass et al. [24] and stemming from the third prior information, we propose a new energy term to force the curvature of the snake to be similar to that of the prior shape. This term assesses the normalized curvature difference between the prior-shape curve and the active-contour curve. As a result, the snake evolution is smooth and converges rapidly towards the true boundaries of the foot by considering the prior shape and the gradient information of the image.

Choosing the initial contour is an important issue in active-contour methods. The initial contour is the downsized prior shape, which is vertically placed inside the plantar foot surface. Thus, the snake avoids false boundaries when evolving. Other prior information allows us to speed up the method as described in Section 2. As preliminary results, we demonstrate that blind-segmentation methods, which do not include prior information, cannot segment our infrared plantar foot images.

We compare the new method with the prior-shape-based snakes proposed by Ahmed et al. [39] and Chen et al. [38]. These two methods have been selected because they are both based on a snake model and incorporate prior-shape information. The prior-shape energy term is the Euclidean distance between the snake curve and that of the prior shape in the Chen et al. method [38].

This distance is calculated in the Fourier space regarding the Ahmed et al. method [39]. Here, we add a new prior-shape energy term to the energy equation of the snake that minimizes the curvature difference between the prior-shape curve and the active-contour one. We expect that this new approach will be more efficient for the segmentation of the plantar foot in thermal images as we know that feet contours present only soft curvatures.

All of the tested methods benefit from the same efficient initialization strategy described above. Our aim is to qualitatively and quantitatively characterize the performance of the proposed method when compared to these two methods for plantar foot thermal image segmentation. As we intend to develop smartphone applications, this method has to be robust and fast. We evaluate the robustness of the three methods to initial contour position variations and when noise is present. The processing time is assessed to evaluate the rapidity of the methods. Finally, a pilot clinical study is presented.

The remainder of the paper is organized as follows. Section 2 presents the segmentation methods using prior-shape snakes: we first gives a brief review of the approaches of Ahmed et al. [39] and Chen et al. [38] and describe the proposed method. In Section 3, we justify the choice of the thermal camera Flir One Pro, describe the acquisition protocol and give details about the database construction and preprocessing. In Section 4, we present our preliminary results, and some blind-segmentation methods are applied to our thermal images. Then, the three prior-shape snake-based methods are applied to our database images. In Section 5, the proposed method is used for the detection of hyperthermia in DF patients. Finally, our conclusions and perspectives are presented in the last section.

## 2. Methods

Starting from an initial contour, the active contour *C* evolves through time to minimize a total energy function, which is a combination of internal and external energies. Prior-shape active contours add an extra energy term, namely a distance between the active contour *C* and $C^{*}$, the prior-shape contour. Since we are searching for a single closed contour, snakes are chosen in this study instead of a level-set formulation. Two existing prior-shape snakes are first briefly described, that of Ahmed et al. and that of Chen et al. We then present the new method.

### 2.1. Fourier-Based Prior-Shape Snake

The method developed by Ahmed et al. [39] is a Fourier-based snake with a greedy implementation [40]. The total energy function of the snake is the one proposed by Kass et al. [24]:(1)$$E_{Total}\left(C\right)\;=\;\int_{0}^{1}\left(E_{intern}\left(C\right)\;+\;E_{image}\left(C\right)\;+\;E_{con}\left(C\right)\right)ds,$$
where an element *C(s,t) = (x(s),y(s))* along the contour depends on the curvilinear abscissa *s* ∈ [0, 1] and on time *t*; *x* and *y* are the pixel coordinates. The internal energy $E_{intern}$ contains two terms: length and curvature. The image energy $E_{image}$ is derived from the gradient information. The external constraint *$E_{con}$* is the balloon energy [41] that forces the movement of the snake towards the object boundaries. To consider the prior shape, an extra term $E_{PS}$ is added to the total energy function of the snake:(2)$$\begin{array}{cc} E_{Total}\left(C\right) & \;=\;\int_{0}^{1}\left(E_{intern}\left(C\right)\;+\;E_{image}\left(C\right)\;+\;E_{con}\left(C\right)\;+\;E_{PS}\left(C\right)\right)ds. \end{array}$$

This term assesses the shape matching performed directly in the Fourier domain. Let $C_{k}\;=\;x_{k}\;+\;j\cdot y_{k}$ be the complex coordinates of one point *k* of the *n* points belonging to *C*. The discrete Fourier transform of *C* leads to a set of Fourier coefficients:(3)$$Z_{k}\;=\;R_{k}e^{\left(j\theta_{k}\right)},k\;=\;-\frac{n}{2},\ldots ,\frac{n}{2}-1,$$
where $R_{k}$ is the amplitude, and $\theta_{k}$ is the phase. The normalized Fourier descriptors are noted ${\hat{Z}}_{k}\;=\;{\hat{R}}_{k}e^{\left(j{\hat{\theta }}_{k}\right)}$, where
(4)$${\hat{Z}}_{k}\;=\;0,\left(k\;=\;0\right),$$
(5)$${\hat{R}}_{k}\;=\;\frac{R_{k}}{R_{1}},\left(k\neq 0\right),$$
(6)$${\hat{\theta }}_{k}\;=\;\theta_{k}-\theta_{1},\left(k\neq 0\right).$$

The extra prior-shape energy is written:(7)$$E_{PS}\;=\;\gamma \sqrt{\sum_{k\;=\;-\frac{n}{2}}^{\left(\frac{n}{2}-1\right)}{\left|{\hat{Z}}_{k}-{\hat{Z}}_{k}^{*}\right|}^{2}}.$$

${\hat{Z}}_{k}$ and ${\hat{Z}}_{k}^{*}$ are the normalized descriptors of the curve *C* and $C^{*}$, respectively. During the evolution of the active contour, the prior shape needs to be iteratively adjusted to the evolving snake by a scale factor, a rotation angle and a translation parameter. The normalization of the descriptors (Equations (Equation 5) and (Equation 6)) avoids updating the scaling and also the rotation between the snake and the prior shape.

Setting the first Fourier descriptor to zero (Equation (Equation 4)) avoids updating the translation factor. The greedy minimization process differs from the original version of Kass et al. At each iteration, the neighborhood of each point on the curve is examined, and the point in the neighborhood that minimizes the energy is chosen as the new location.

### 2.2. Prior Shape-Based Geodesic Snake

Chen et al. [38] presented a prior-shape snake method based on the geodesic energy of Caselles et al. [42]:(8)$$E_{Total}\;=\;\int_{0}^{1}g(|\nabla I|\left(C\right))|C_{s}|ds,$$
where the curve *C*= (*x*(*s*), *y*(*s*)), $C_{s}\;=\;\frac{\partial C}{\partial s}$, the gradient of image I is defined as $g\left(|\nabla I|\right)\;=\;\frac{1}{\left(1\;+\;b|\nabla G_{\sigma }*I{|}^{2}\right)}$, *b* ≥ 0 is a parameter, and $G_{\sigma }$ is a Gaussian kernel with standard deviation σ. The authors proposed finding the curve *C* and the transformations μ, *R* and *T* (the scale, the rotation matrix of an angle θ and the translation parameters, respectively) such that the prior-shape $C^{*}$ matches the curve *μR(θ)C + T*. An extra energy term is added in the basic energy equation to consider the prior-shape information:(9)$$E_{Total}\left(C,\mu ,R,T\right)\;=\;\int_{0}^{1}(g(|\nabla I|\left(C\right)\;+\;\frac{\lambda }{2}d^{2}\left(\mu RC\;+\;T\right))|C_{s}|ds,$$
where *λ ≥ 0* is a weight, and *d(x,y)=d($C^{*}$,(x,y))* is the distance estimated by fast marching [43] between the point *(x, y)* on the image *I* and the curve $C^{*}$. The curve *C* and the transformation parameters μ, *R* and *T* evolve in such a way as to minimize Equation (Equation 9). Gradient descent minimization was conducted to find the transformation μ, *R* and *T* that maps *C* to $C^{*}$. The gradient/variation descent for the energy and transformation parameters is performed as follows:(10)$$\frac{\partial C}{\partial t}\;=\;-\left[\nabla g\cdot \eta \;+\;gk\;+\;\lambda \mu \left(d\nabla d\right)\cdot \left(R\eta \right)\;+\;\lambda d^{2}k\right]\eta ,$$
(11)$$\frac{\partial \mu }{\partial t}\;=\;-\lambda \int d\nabla d\cdot RC\left|C_{s}\left(s\right)\right|ds,\mu \left(0\right)\;=\;\mu_{0},$$
(12)$$\frac{\partial \theta }{\partial t}\;=\;-\lambda \mu \int d\nabla d\cdot \frac{\partial R}{\partial \theta }C)|C_{s}\left(s\right)|ds,\theta \left(0\right)\;=\;\theta_{0},$$
(13)$$\frac{\partial T}{\partial t}\;=\;-\lambda \int d\nabla d\left|C_{s}\left(s\right)\right|ds,T\left(0\right)\;=\;T_{0},$$
where η is the outward unit normal to *C*, and *k* is the curvature of the curve *C*. The functions *g*, *k* and η are evaluated at *C*, while *d* is evaluated at *μRC + T*.

### 2.3. The Proposed Prior-Shape Snake

As already mentioned, the aim of this study is to include all the prior information about the thermal plantar foot images to be segmented:The plantar foot contour to be found is a single closed contour.The average shape of the plantar foot is known.The plantar foot contour and the average shape contour present both soft curvatures.No strong edges are present inside the plantar foot region while strong ones appear elsewhere in the image. Some true boundaries can be hidden.The orientation of the foot is known (vertical or quasi-vertical).

The general strategy we propose is the following. As we are expecting a single closed contour, a snake-based method is preferred. We propose to use the snake of Kass et al. [24] whose energy function is described in Equation (Equation 1) and to modify it by adding the prior-shape energy $E_{PS}$. This energy function stems from the third assumption (only soft curvature should be present). It assesses the normalized difference between the curve curvature and the prior-shape curvature during the contour evolution.

Choosing the initial contour is an important issue in active-contour methods. To increase the efficiency of the method, the initial contour is chosen to be placed inside the foot region as close as possible to the true boundaries of the foot. The downsized prior-shape initial contour, which is oriented vertically is placed at the gravity center of the plantar foot surface. As a result, the snake avoids false boundaries present outside the plantar foot surface, when evolving.

As a result, we expect that the snake evolving from the initial contour (placed inside the foot region and presenting soft curvatures) converges rapidly and efficiently towards the true boundaries of the foot. We now give details about the various energies of Equation (Equation 1) as well as the prior-shape energy. Kass et al.’s energy terms $E_{intern}$, $E_{image}$ and $E_{con}$ in Equation (Equation 1) are given by the following expressions:(14)$$\begin{array}{cc} E_{intern} & \;=\;\alpha |C_{s}{\left(s\right)|}^{2}\;+\;\beta {\left|C_{ss}\left(s\right)\right|}^{2}, \end{array}$$(15)$$\begin{array}{cc} E_{image} & \;=\;-W_{edge}{\left|\nabla I\left(C\right)\right|}^{2}, \end{array}$$(16)$$\begin{array}{cc} E_{con} & \;=\;{\delta |\eta \left(C\right)|}^{2}, \end{array}$$
where $C_{s}\;=\;\frac{\partial C}{\partial s}$, $C_{ss}\;=\;\frac{\partial^{2}C}{\partial s^{2}}$, and η(*C*) is the outward unit norm to the curve *C*. The parameters α and β control the internal energy, whereas $E_{image}$ and $E_{con}$ depend on the parameters $W_{edge}$ and δ, respectively. As the initial contour has to grow, we choose δ to be a positive constant. The prior-shape energy function is the weighted normalized difference between the curvature of the snake $C_{ss}$ and the prior-shape curvature $C_{ss}^{*}$.
(17)$$E_{PS}\;=\;\gamma {\left|C_{ss}\left(s\right)-\zeta C_{ss}^{*}\left(s\right)\right|}^{2},$$
where γ is the weight of the prior-shape energy, and $\zeta \;=\;\frac{|C_{ss}\left(s\right)|}{|C_{ss}^{*}\left(s\right)|}$ is the normalization factor. Minimizing the total energy function $E_{Total}$ (Equation (Equation 2)) leads to solving the Euler–Lagrange equation:(18)$$-\alpha C_{ss}\;+\;\left(\beta \;+\;\gamma \right)C_{ssss}-\gamma \zeta C_{ssss}^{*}\;+\;\frac{\partial (E_{image}\;+\;E_{con})}{\partial C}\;=\;0,$$
where $C_{ssss}\;=\;\frac{\partial^{4}C}{\partial s^{4}}$ and $C_{ssss}^{*}\;=\;\frac{\partial^{4}C^{*}}{\partial s^{4}}$. During the evolution of the active contour, the prior shape needs to be iteratively adjusted to the evolving snake by a scale factor μ, a rotation angle θ and a translation parameter *T*. To speed up the process, our method does not require such adjustments. During the snake evolution, both snake and prior shape are examined from the same initial point (the lowest point of the calcaneus). As a result, the method is invariant to rotation. The prior-shape energy is a function of the relative positions of *C* and $C^{*}$ (curvature difference), which avoids updating the translation parameter. Lastly, we used normalized curvatures, the ζ term in Equation (Equation 17), which avoids updating the scale factor.

## 3. Materials and Preprocessing of the Data

In this section, we detail the infrared camera and describe the recruitment protocol, the acquisition protocol and the preprocessing of the plantar foot thermal images.

### 3.1. Choice of the Infrared Camera

Three parameters were taken into consideration for choosing the smartphone IR camera:

#### 3.1.1. Resolution

The two plantar feet, vertically oriented, must be included in the image with an external margin of 5 cm from the border of the image. The feet must be separated by 10 cm from each other so that the image can be divided into two equivalent images. In the horizontal orientation, 40 cm is sufficient to contain the width of both feet, including a 5 cm margin and 10 cm separation between the feet. In the other dimension, the largest foot we consider is 30 cm. The field of view is then 40 × 40 cm^2^. For the medical application that we have in mind, the smallest hyperthermia region we consider is a circle of approximately 1 cm in diameter (a toe for example). According to the Shannon theorem, at least 2 pixels are needed to describe this object. This means that any camera with a resolution greater than 80 × 80 pixels is suitable.

#### 3.1.2. Sensitivity

A point-to-point difference between the right and left foot above 2.2 °C is considered as hyperthermia. Thus, a sensitivity of 0.2 C is enough for the camera to detect these possible variations of interest.

#### 3.1.3. Spectral Range

The average skin temperature of a healthy person in normal conditions is 32 °C. According to the Wien law [44], this is related to a peak wavelength of 9.5 μm. This corresponds to the lowest infrared zone.

The chosen camera is the FlirOne Pro thermal camera Figure 2 designed to be plugged into a smartphone. This camera has a resolution of 160 × 120 pixels and a spectral range of 8–14 μm. The FlirOne Pro can detect temperature differences of 0.1 °C.

### 3.2. Database and Acquisition Protocol

The acquisition campaign was conducted in the PRISME laboratory of the University of Orleans in January 2017 for two weeks. A total of 25 healthy (non-diabetic) persons participated in our acquisition campaign. This sample group was composed of 10 women and 15 men with a mean age of 34 years old from staff members of the laboratory and the University of Orleans, France. The acquisition took place in a room with an average temperature of 20 °C and a controlled luminosity.

We included any non-diabetic person from the staff of the university who had no foot problems. Each participant was first asked to read and sign the informed consent form and fill in the Material Safety Data Sheet containing personal information. The participants were requested to remove their shoes and socks. After a 15 min interval to allow the feet to recover their normal temperature, the person laid down on a stretcher and placed their feet at the end of the stretcher in a vertical position and 10 cm apart. The thermal picture was taken freehand using a Samsung Galaxy S8 and a Flir One Pro camera. The operator held the camera at about 1 m from the plantar feet. An example of a typical acquired image is shown in Figure 3a.

### 3.3. Preprocessing of the Data

Images from the Flir One Pro camera are in JPEG format of 480 × 640 pixels. Labels are displayed at the bottom, and a temperature bar is displayed on the right side of the image (Figure 3a). For the 25 images of our database, the temperature bar and the labels are first removed from the image. Then, the left foot is separated from the right foot by splitting the image into two equal parts as shown in Figure 3b,c. The left foot is flipped resulting in the image of Figure 3d. Figure 3b,d are the images to be segmented. The database contains 50 images (right and flipped left foot images of the 25 controls).

## 4. Results and Discussion

In this section, we present the preliminary results, segmenting the images of interest using blind methods (with no prior shape). We then test the proposed method and the two classical prior snake methods (that of Ahmed et al. and that of Chen et al.) on the images of our database composed of 50 images coming from 25 healthy persons (10 women and 15 men with a mean age of 34 years old).

### 4.1. Segmentation Using Blind Methods

We first apply on one image of the database (results are equivalent for other images) the state-of-the-art segmentation methods that are not based on prior-shape information (blind segmentation). The results are depicted in Figure 4. The red curve corresponds to the contour found by various methods, while the green curve represents the ground truth contour manually given by an expert.

The segmentation given by the region-growing method [25] is shown in Figure 4b. Figure 4c,d shows the results of the classical snake method of Kass et al. [24] and the Chan–Vese method [23]. We also applied other sophisticated methods based on the level set, such as the ones suggested in [26,27,28] as shown in Figure 4e–g). Finally, Figure 4h presents the results given by the graph cut-based method proposed in [29].

From the analysis of Figure 4, it can be seen that the blind-segmentation methods fail to segment the plantar foot thermal images since strong false boundaries are present in the image. The methods are strongly sensitive to this noise, and it is expected that prior-shape methods could overcome these difficulties. The following section presents tests of the three prior-shape snakes on plantar foot thermal images.

**Figure 4 sensors-22-03835-f004:**
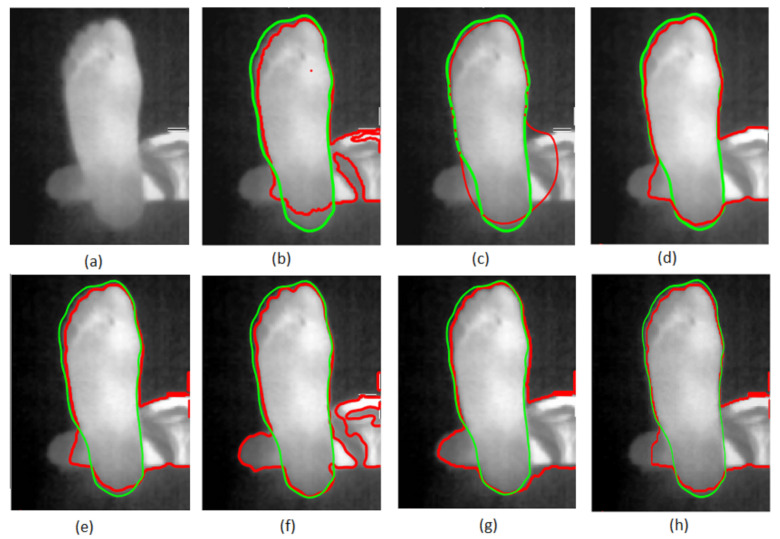
Segmentation of an IR thermal image with seven blind-segmentation methods: (**a**) a typical image, (**b**) region-growing method [25], (**c**) snake method [24], (**d**) Chan and Vese method [23], (**e**) Zhang et al. method [26], (**f**) Dong et al. method [27], (**g**) Li et al. method [28] and (**h**) Ben Salah et al. method [29]. Red curve corresponds to the segmentation results and green one is the ground truth contour.

### 4.2. Segmentation with Prior-Shape Active-Contour Methods

First, we present the construction of the prior shape and the initialization procedure. The qualitative results are shown first, followed by the quantitative ones. Then, the robustness and speed are also evaluated.

#### 4.2.1. The Prior Shape and Initialization Procedure

The method used to obtain the prior shape is based on that proposed in [38]. Ten thermal foot images randomly extracted from the database were first manually segmented by an expert. They were registered regarding their gravity centers (Figure 5a). The gravity center of each foot was calculated by binarizing the image with an adaptive threshold using the Otsu method [45].

Two morphology operations were applied to the binary image: an erosion by a structuring element that has the form of a line followed by a dilation by a square structuring element. The prior shape is the average of these ten curves as shown in Figure 5b. The prior shape is downsized by a scale factor of 2 and is placed vertically at the gravity center of the foot surface as displayed in Figure 5c. This prior shape and initialization procedure were applied to all the methods tested below.

#### 4.2.2. Qualitative Results

We tested the proposed method and the two classical prior snake methods (that of Ahmed et al. and that of Chen et al.) on plantar foot thermal images. After several trials, the parameters of each method were selected to provide the best possible results. The parameters of our method were α = 0.1, β = 4, $W_{edge}$ = 20, δ = 0.2, and γ = 35. For the method of Ahmed et al., the parameters were α = 1.2, β = 0.1, $W_{edge}$ = 5, δ = 0.1, and $\gamma_{0}$ = 0.5.

Then, the parameters and initial transformation for the Chen method were α = 0.2, *b* = 0.1, σ = 0.3, γ = 8, $\mu_{0}$ = 1, $\theta_{0}$ = 0, and $T_{0}$ = [0; 0]. We first qualitatively evaluated the performance of the three tested methods. The results are presented in Figure 6 for three typical images selected from the database. The first row corresponds to our method Figure 6a–c, the second row shows the method of Ahmed et al. Figure 6d–f, while that of Chen et al. is presented in the last row Figure 6g–i. The blue curve corresponds to the contour found by the methods, while the green curve represents the ground truth contour manually assessed by an expert.

The results show that prior-shape-based snake segmentation is a potential candidate to segment the infrared images of our database. Moreover, the proposed methods appeared to give closer final contours to the ground truth contour. A quantitative evaluation is thus needed to confirm these observations.

#### 4.2.3. Quantitative Results

Quantitative results were assessed from the 50 images of the database. Two metrics were used. The first is the root-mean-square error (RMSE) between the ground truth contour given by the expert and the final contours given by the three methods. At each point of the ground truth contour, the closest point of the contour is estimated to calculate the final RMSE. The second metric is the Dice Similarity Coefficient (DSC) [46] quality index. This score assesses the similarity between the region given by the ground truth contour and the the results found with the three methods.

Table 1 shows the average scores (mean) given by the methods with their respective standard deviations (STD). In all of the above tables, the best results are in bold font. A unilateral Student’s *t*-test with the null hypothesis such that mean 1 is equal to mean 2 for two populations and with alternative hypothesis such that mean 1 is greater than mean 2 was performed with a level of significance of 5% (Table 2). If the *t*-test value is greater than 1.65, then the null hypothesis is rejected, meaning that our method performs better than the other methods.

Based on the two tables, we see that the proposed method outperformed the two others in terms of RMSE error (5.12 ± 1.88) and DSC quality index (0.940 ± 0.02). This result is significant based on the *t*-test values, which are all greater than 1.65.

**Table 1 sensors-22-03835-t001:** RMSE and DSC of the segmentation methods.

	RMSE (Mean ± STD)	DSC (Mean ± STD)
Our method	**5.12 ± 1.88**	**0.940 ± 0.02**
Ahmed et al. [39]	10.99 ± 4.24	0.855 ± 0.07
Chen et al. [38]	6.16 ± 1.38	0.930 ± 0.02

**Table 2 sensors-22-03835-t002:** Student’s *t*-test values between the proposed method and the two other methods.

	Our/Ahmed et al. [39]	Our/Chen et al. [38]
RMSE	5.87	2.95
DICE	8.20	2.34

#### 4.2.4. Robustness and Speed

The robustness of the three methods to initial contour variations was tested by changing its position as illustrated in Figure 7a–c. Our method is in row 2 of Figure 7d–f, the Ahmed et al. method is in row 3 of Figure 7g–i, and the Chen et al. method is in the fourth row of Figure 7j–l. The blue and green curves present the final contours given by the tested method and the ground truth contour, respectively.

The segmentation result given by the proposed method is almost invariant to the position of the initial contour modification, while the Ahmed et al. and Chen et al. methods are more sensitive to this perturbation. This observation was confirmed by the quantitative results as presented in Table 3 for the 50 images of the database. The first column of the table presents the three different positions of the initial contour in Figure 7a–c.

The DSC and RMSE scores given by the three different methods are presented as indicated in the table. We can see that the methods proposed by Ahmed et al. and Chen et al. are sensitive to the variation of the initial contour, whereas our method is robust and gave better DSC and RMSE scores.

**Figure 7 sensors-22-03835-f007:**
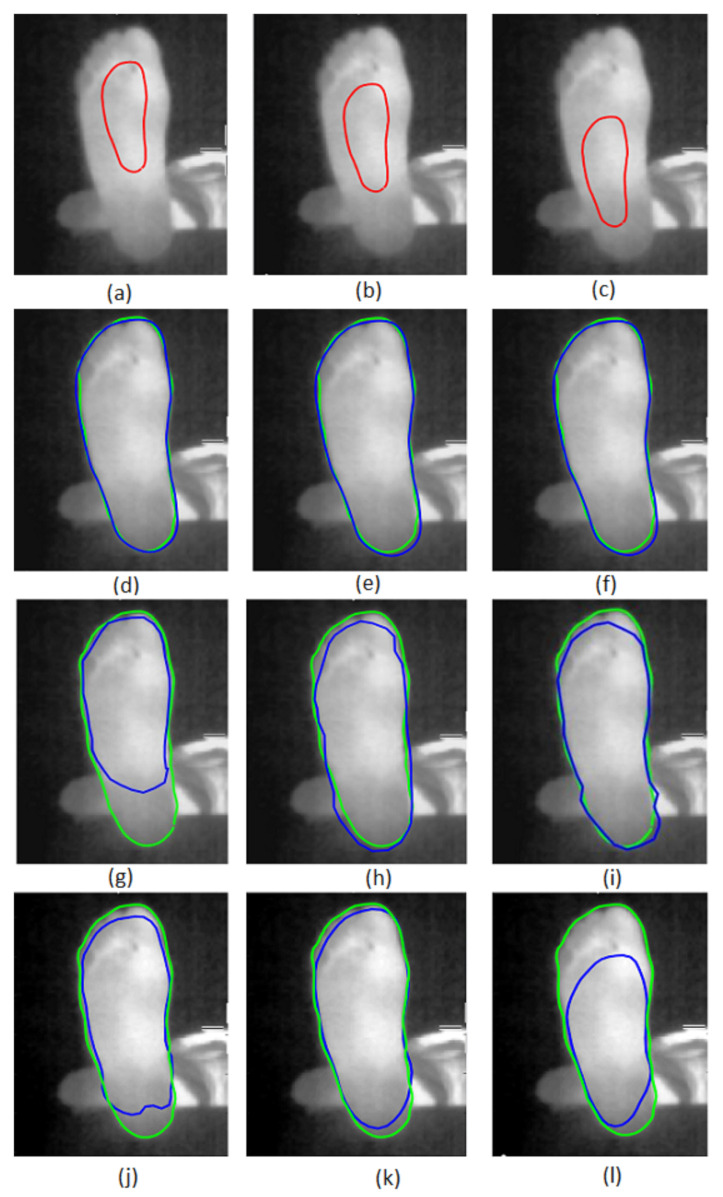
Segmentation of a plantar foot thermal image with different initial contours. The different positions of the initial contour are in (**a**–**c**). Our method’s results are in (**d**–**f**). The Ahmed et al. [39] method is (**g**–**i**), and the Chen et al. [38] method is (**j**–**l**). Blue and green curves correspond respectively to segmentation results and the ground truth contours.

**Table 3 sensors-22-03835-t003:** The RMSE and DSC of the segmentation methods with different initial contour positions.

Initial Contour Position	Our Method		Ahmed et al. [39]		Chen et al. [38]	
	DSC	RMSE	DSC	RMSE	DSC	RMSE
position 1	**0.936**	**5.90**	0.802	14.11	0.913	8.21
position 2	**0.940**	**5.12**	0.855	10.99	0.930	6.16
position 3	**0.932**	**6.60**	0.781	17.90	0.882	9.82

We also evaluated the robustness of the three methods to different additive noise levels. To this end, we added to the image a zero-mean Gaussian white noise with different variances. The average values of DSC and RMSE were calculated for our images and are presented in Table 4. The different signal-to-noise ratios are indicated in the first column. The DSC and RMSE mean values of the proposed method, Ahmed et al. method and Chen et al. method are presented in columns 1, 2 and 3, respectively.

**Table 4 sensors-22-03835-t004:** Robustness to an additive Gaussian noise with different signal-to-noise ratios (SNR).

SNR (dB)	Our Method		Ahmed et al. [39]		Chen et al. [38]	
	DSC	RMSE	DSC	RMSE	DSC	RMSE
25	**0.938**	**5.45**	0.842	11.07	0.921	6.57
15	**0.935**	**6.88**	0.840	11.19	0.911	8.08
10	**0.927**	**6.01**	0.838	11.57	0.893	9.65
5	**0.920**	**7.51**	0.829	12.07	0.881	9.78

From this table, one can see that the proposed method and the methods of Ahmed et al. are less sensitive to noise compared to the Chen et al. method. As a conclusion, the proposed method is robust to position variations of the initial contour and also robust to additive noise.

An important criterion is the processing time, especially for a smartphone deployment. To that end, we measured the average CPU time on a Dell Precision Station 1700 (i7-4790 Core and CPU frequency of 3.60 GHz). Table 5 summarizes the results obtained on the database of thermal images whose dimensions are 257 × 385 pixels. This table indicates that the method of Ahmed et al. performed better than the other two methods. Our method ranked second with a speed difference of less than 1 s. The CPU time of our method is 2.3 s, which is suitable for a smartphone application.

**Table 5 sensors-22-03835-t005:** Average CPU times obtained with the different methods.

Our Method	Ahmed et al. [39]	Chen et al. [38]
2.3 s	**1.5 s**	6.5 s

To conclude, we performed an evaluation of the segmentation methods on our database of 50 thermal images coming from 25 non-diabetic persons (15 men and 10 women with a mean age of 34 years). First, the blind-segmentation methods failed to segment such plantar foot thermal images since strong false boundaries are present in the image. Second, we compared our proposed prior-shape-based snake method to classical prior-shape snakes, namely, the methods of Ahmed et al. and Chen et al. The proposed method outperformed the other two tested methods in terms of the segmentation quality. In addition, it is fast and robust to noise and initial condition variations. It can therefore be considered for smartphone-based early diagnosis of DF or for DF ulcer prevention and monitoring.

## 5. Hyperthermia Detection

The objective of this section is to demonstrate the clinical potential of this approach. This pilot clinical study illustrates the interest of the proposed segmentation method for hyperthermia detection in diabetic foot (DF) problems. Hyperthermia is defined as a temperature difference greater than 2.2 °C between a foot region and the same region on the contralateral foot. This information is one of the most promising indicators for foot ulcer prevention in DF [4]. To precisely visualize and analyze this important information, we propose to use plantar foot thermal images.

These images were acquired freehand using a Samsung Galaxy S8 and the Flir One Pro camera. We first automatically segmented them using the proposed segmentation method (no manual intervention was needed). We then registered the two segmented feet images using the iterative closest point registration method [47]. Finally, the point-to-point absolute temperature difference image |ΔT| was calculated. The same thermal scale was used for all the |ΔT| difference maps allowing a better visual comparison. The case of two control subjects (no diabetes) in Figure 8a,b was first analyzed, and |ΔT| is displayed in Figure 8c,d.

It is known that the average |ΔT| is about 1 °C for healthy subjects. The average |ΔT| is 0.27 °C for the first person and 0.44 °C for the second in agreement with the expected values since both persons do not have diabetic foot. No hyperthermia is detected. Figure 9 shows the same study for two DF patients with ulcers. The first person has an ulcer on the left big toe as shown in Figure 9a. The average |ΔT| (Figure 9c) is 1.28 °C, which is well above the 1 °C for healthy subjects. The |ΔT| on the left big toe is about 3 °C, much higher than the 2.2 °C required to decide whether or not hyperthermia is present.

The second case is a DF patient who has an ulcer under the toes as shown in Figure 9b. The |ΔT| mean value (Figure 9d) is 1.57 °C and |ΔT| is 3.5 °C in this region. The new tool that we propose here is a user-friendly mobile technology for the precise visualization and analysis of the plantar foot temperature in DF.

## 6. Conclusions and Perspectives

In this paper, we presented a new prior-shape snake segmentation method for plantar foot thermal images. This method was designed to include all of the prior information regarding the images of interest. The initial contour was the downsized prior shape placed inside the plantar foot surface in such a way that the snake avoids false boundaries when evolving. We introduced a new term into the snake of Kass et al. to force the curvature of the moving snake to be similar to that of the prior shape. As a result, the snake evolution produced a smooth contour that rapidly converged towards the true boundaries of the foot.

The proposed method was compared to two prior-shape snake methods: the Ahmed et al. and the Chen et al. methods. The proposed method, Ahmed et al.’s method and Chen et al.’s method were tested on the images of our database, which was composed of 50 images coming from 25 healthy persons who were 10 women and 15 men with a mean age of 34 years old from staff members of Orleans University, France. The results showed the superiority of our method in terms of the RMSE (5.12 pixels) and DSC (94%). Our method converged rapidly toward the optimal solution, and it was less sensitive to initial contour variations and to noise.

Nevertheless, the proposed method still has certain limitations, which open the way for new studies. The first limitation is related to the choice of the initial parameters of the method. We are, at the present time, working on a new way to automatically choose these parameters. The case of a severely amputated foot, where the contour of the foot strongly varies from one patient to another, is similarly a limitation of the method. Our database is not yet large enough, and we are currently performing an acquisition campaign in several hospitals in order to obtain a large database. We intend to test deep convolutional neural network methods.

Finally, the proposed approach was implemented in a mobile and user-friendly application for the analysis of plantar foot temperature in DF. This application was designed to detect regions where the point-to-point temperature difference between both feet is greater than 2.2 °C to recognize hypethermia [4]. No constraining isolation system was needed, images were taken freehand with a smartphone equipped with a dedicated thermal camera, and the fully automatic processing of the data was performed immediately in the smartphone itself. We expect that this overall protocol can be generalized in clinical routines or even at home for DF patients.

## Figures and Tables

**Figure 1 sensors-22-03835-f001:**
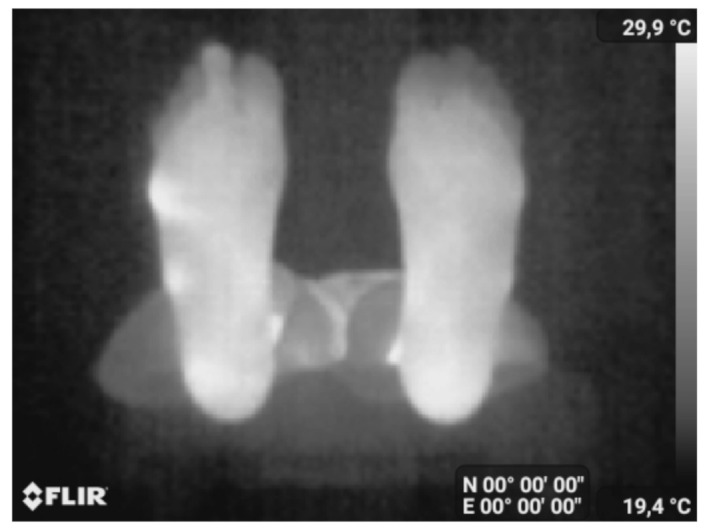
Example of an acquired thermal image.

**Figure 2 sensors-22-03835-f002:**
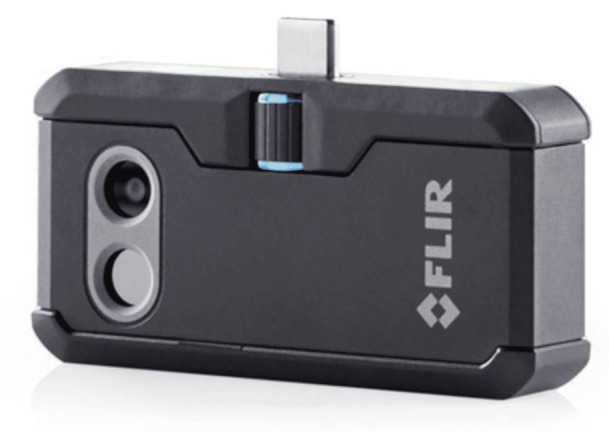
The FlirOne Pro camera.

**Figure 3 sensors-22-03835-f003:**
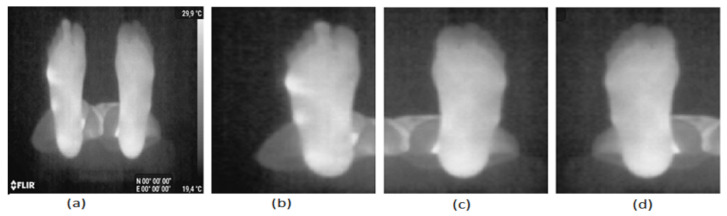
Acquired image preprocessing, (**a**) the two feet, (**b**) the right foot, (**c**) the left foot and (**d**) the flipped left foot.

**Figure 5 sensors-22-03835-f005:**
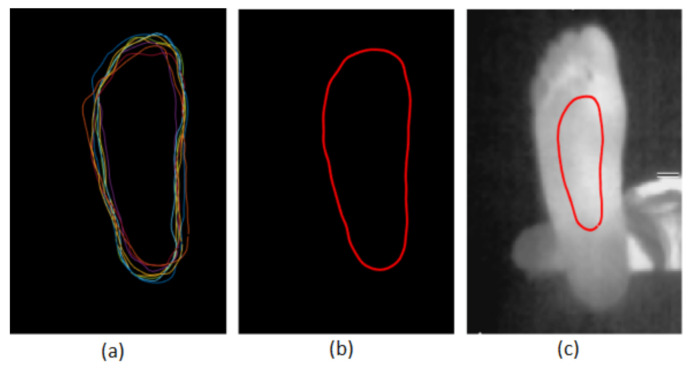
(**a**) Ten registered foot contours. (**b**) The prior-shape contour. (**c**) The initial contour.

**Figure 6 sensors-22-03835-f006:**
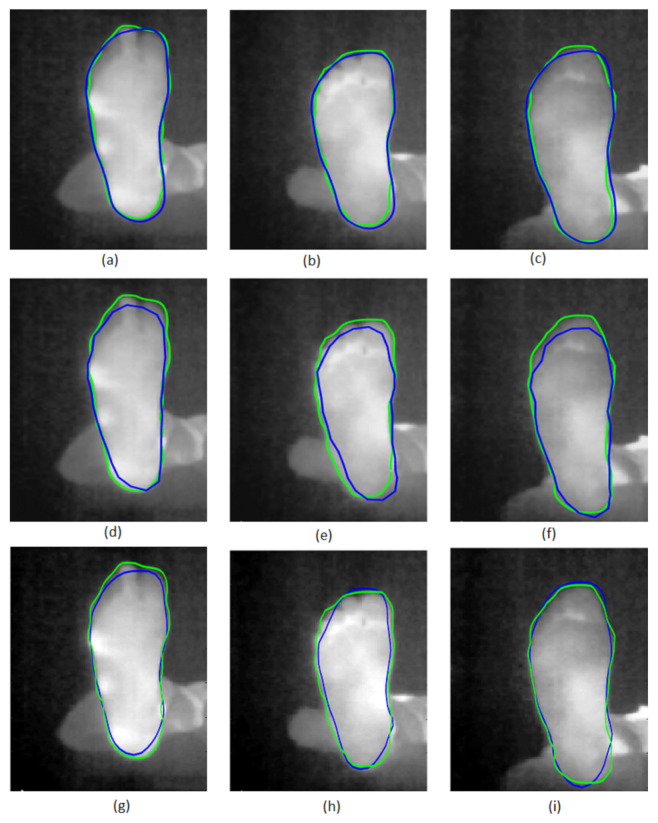
Segmentation of three plantar foot thermal images. The proposed method results are in (**a**–**c**). The Ahmed et al. results are in (**d**–**f**). The Chen et al. results are in (**g**–**i**). Blue and green curves correspond respectively to segmentation results and the ground truth contours.

**Figure 8 sensors-22-03835-f008:**
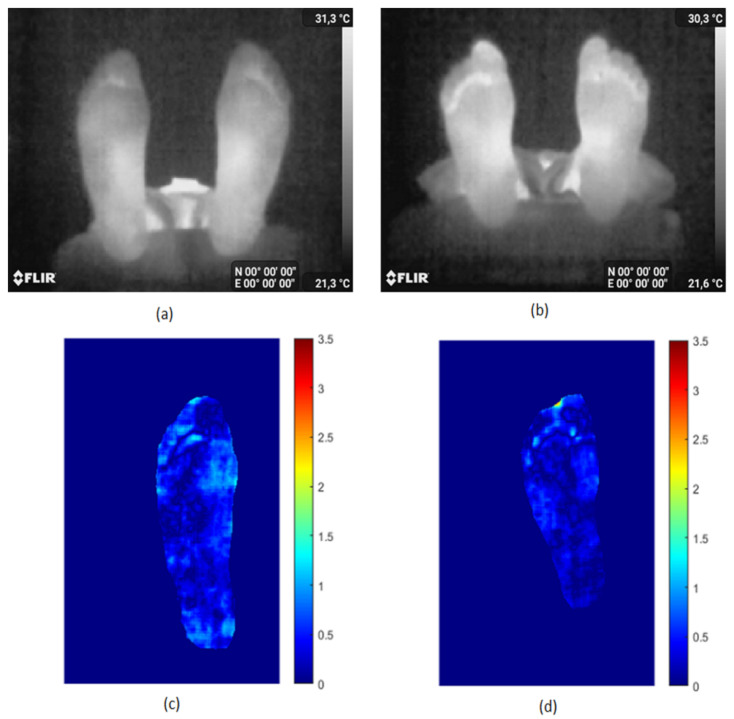
Thermal difference map |ΔT| for two control subjects.images (**a**,**b**) correspond to two healthy persons. Their thermal maps |ΔT| are respectively in (**c**,**d**).

**Figure 9 sensors-22-03835-f009:**
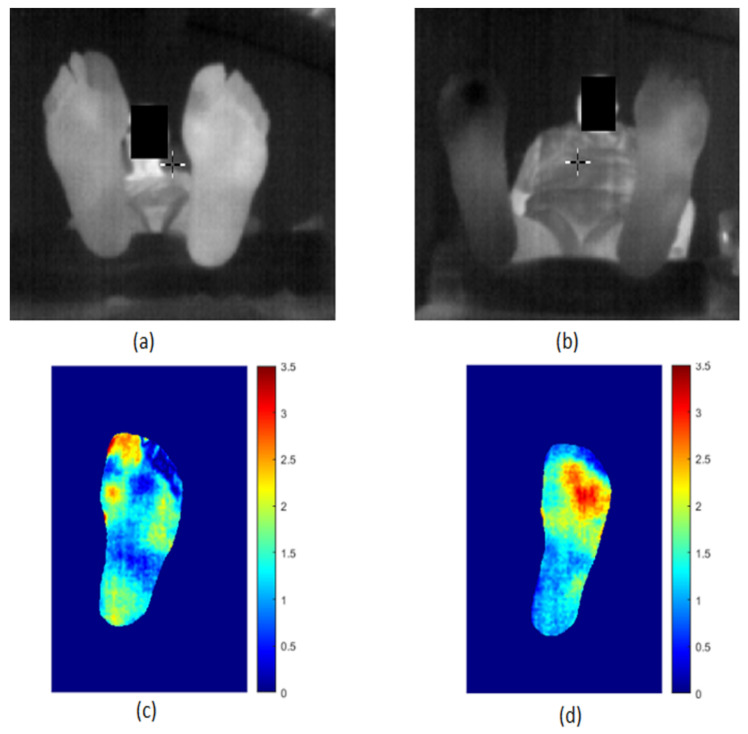
Thermal ifference map |ΔT| for two DF subjects with ulcers. images (**a**,**b**) correspond to two diabetic subjects with ulcers. Their thermal maps |ΔT| are respectively in (**c**,**d**).

## Data Availability

Images are currently only available internally to our partners of the STANDUP #777661 project. It is also worth mentioning that the database will be publicly available on https://www.standupproject.eu/home/fr (accessed on 11 April 2022) once the research project is achieved on 30 June 2023.

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
