# Peer review of "Segmentation of Plantar Foot Thermal Images Using Prior Information"

_sensors, 2022, doi:10.3390/s22103835_

Round 1

Reviewer 1 Report

In the abstract, the significance of the research should be given, and the steps of the method you proposed should be described specifically.

The first paragraph of the introduction should be simplified to clarify the significance of the research.

I don't think that part 3.1 is the right place in the results.

Some references are old, and replacement is recommended.

Author Response

Authors’ Response to the Review Comments

Journal: Sensors SI Contactless Sensors for Healthcare

Manuscript #: sensors-1702021

Title of the Paper:  Segmentation of plantar foot thermal images using prior information

Authors: Asma Bougrine, Rachid Harba, Raphael Canals, Roger Ledee, Meryem Jabloun, Alain Villeneuve

We would like first to thank the handling editor and the anonymous reviewers for their constructive comments and reviews. We have carefully addressed and clarified each point raised by the editor and the reviewers. Reviewer’s comments are in blue font while and our responses are in black font. Revisions in the manuscript are marked up using the “Track Changes” function de LaTeX. In the manuscript, modifications asked by Reviewer#1 are in underlined blue font and those of Reviewer#3 are in underlined green font.

Reviewer#1

Concern # 1: In the abstract, the significance of the research should be given, and the steps of the method you proposed should be described specifically.

Author response: We want to thank the reviewer for this comment.

Author action: We updated the abstract by adding more details about the significance of our study and reformulated it to highlight the steps of the proposed method.

Concern # 2: The first paragraph of the introduction should be simplified to clarify the significance of the research.

Author response: Thank you for this comment. We agree with you that the first paragraph needs to be simplified to express the significance and interest of the study.

Author action: We simplified the first paragraph of the introduction to express better the significance of our study.

Concern # 3: I don't think that part 3.1 is the right place in the results.

 Author response: Thanks for bringing this to our attention your suggestion is appropriate.

Author action: We updated the manuscript by placing the part 3.1 in a separate section: 3. Materials and preprocessing of the data.

Concern # 4: Some references are old, and replacement is recommended.

Author response: Thank you for highlighting this point, some references are indeed old because we use basic active contour methods such as the Snake of Kass et al. 1988, the Chan & Vese et al. 1999, Caselles et al. 1997, etc. Even if we think that the presence of the old references is important in this paper, we agree with reviewer#1 that the presence of recent publications is of big interest.

Author action: we went through the sections of the paper, and we added publications that are more recent.

Reviewer 2 Report

The authors present a novel technique to segment the thermogram of the feet from diabetic patients. Even when the technique is based on previous works from other researchers, the technique is interesting for the purpose they proponed. They also present a protocol to acquire the thermogram, which vary slightly from the most used one. The results were compared with other works, showing that the presented technique has better performance. In resume, it is an interesting work.

Author Response

The authors present a novel technique to segment the thermogram of the feet from diabetic patients. Even when the technique is based on previous works from other researchers, the technique is interesting for the purpose they proponed. They also present a protocol to acquire the thermogram, which vary slightly from the most used one. The results were compared with other works, showing that the presented technique has better performance. In resume, it is an interesting work.

Author response: Thank you, we appreciate all your positive remarks about our work.

Reviewer 3 Report

The topic of the presented manuscript is important, relevant, and original. This is a well written manuscript but some issues could be improved. 

  • Introduction seems enough but could achieve a deeper knowledge about state of art. 

  • The research hypothesis are not described
  • There is no info about gender of the sample. Was mix ? Only man ? Woman ?. Please, add info about it and also in discussion and results. 
  • Where and how were the subjects recruited?
  • The chronology is missing.

  • Statistical analysis and results are clear enough. 

  • Discussion section seems confused, needs to be rewritten in some paragraphs. 

Author Response

Authors’ Response to the Review Comments

Journal: Sensors SI Contactless Sensors for Healthcare

Manuscript #: sensors-1702021

Title of the Paper:  Segmentation of plantar foot thermal images using prior information

Authors: Asma Bougrine, Rachid Harba, Raphael Canals, Roger Ledee, Meryem Jabloun, Alain Villeneuve

We would like first to thank the handling editor and the anonymous reviewers for their constructive comments and reviews. We have carefully addressed and clarified each point raised by the editor and the reviewers. Reviewer’s comments are in blue font while and our responses are in black font. Revisions in the manuscript are marked up using the “Track Changes” function de LaTeX. In the manuscript, modifications asked by Reviewer#1 are in underlined blue font and those of Reviewer#3 are in underlined green font.

Reviewer#3

Concern # 1: The topic of the presented manuscript is important, relevant, and original. This is a well-written manuscript, but some issues could be improved. 

Author response: Thank you, it is great to hear that you see promise in our work.

Concern # 2: Introduction seems enough but could achieve a deeper knowledge about state of art. 

Author response: Thank you for this comment.

Author action: We added more state-of-the-art details on active contours with prior information to  improve the introduction.

Concern # 3: The research hypothesis are not described

Author response: Thank you for your comment.

Author action:  We updated the introduction by developing more the hypothesis of our method. Indeed, theses hypotheses are all information we have about the problem that are incorporated in the Snake method such as the singularity of the desired contour, its known shape, and its soft curvature. The gradient information in the infrared image is also characterized, so that no strong contours are observed inside the foot region. Finally, during the acquisition we ask the participants to keep the foot orientation vertical

Concern # 4: There is no info about gender of the sample. Was mix? Only man? Woman?. Please, add info about it and also in discussion and results. 

Author response: Thanks for pointing at this. The information about gender of the sample was mentioned in the paragraph "Database and acquisition protocol". The sample data was composed of 10 women and 15 men with a mean age of 34 from staff members of the PRISME laboratory and the Orleans University, France.

Author action: We modified the manuscript and mentioned information about the gender of the sample in the results and discussions section and in conclusion.

Concern # 5: Where and how were the subjects recruited?

Author response: Thank you for this question. The acquisition took place in university of Orleans, in a room with an average temperature of 20°C and a controlled luminosity. We included any healthy person from the staff of the university, who has two feet.

Author action: We updated the part "Database and acquisition protocol" to give more information about recruitment protocol and inclusion/exclusion criteria.

Concern # 6: The chronology is missing.

Author response: Thank you for this remark. The acquisition campaign was conducted in January 2017 for two weeks.

Author action: We added this information to the part “Database and acquisition protocol”.

Concern # 7 Statistical analysis and results are clear enough. 

Author response: Thank you for this positive comment.

Concern # 8 Discussion section seems confused, needs to be rewritten in some paragraphs. 

Author response: Thank you for your comment. Each part of the results and discussions section should include a paragraph for discussion.

Author action: We rewrote the discussion paragraphs in the “Result and discussions” section to remove confusion.
